# Pandemic buying: Testing a psychological model of over-purchasing and panic buying using data from the United Kingdom and the Republic of Ireland during the early phase of the COVID-19 pandemic

**Richard P. Bentall**[1]*, **Alex Lloyd**[2], **Kate Bennett**[3], **Ryan McKay**[2], **Liam Mason**[4], **Jamie Murphy**[5], **Orla McBride**[5], **Todd K. Hartman**[1], **Jilly Gibson-Miller**[1], **Liat Levita**[1], **Anton P. Martinez**[1], **Thomas V. A. Stocks**[1], **Sarah Butter**[1], **Frédérique Vallières**[6], **Philip Hyland**[7], **Thanos Karatzias**[8], **Mark Shevlin**[5]

**1** University of Sheffield, Sheffield, England, **2** Royal Holloway, University of London, Egham, England, **3** University of Liverpool, Liverpool, England, **4** University College London, London, England, **5** Ulster University, Coleraine, Northern Ireland, **6** Trinity College Dublin, Dublin, Republic of Ireland, **7** Maynooth University, Maynooth, Republic of Ireland, **8** Edinburgh Napier University, Edinburgh, Scotland

\* r.bentall@sheffield.ac.uk

## Abstract

The over-purchasing and hoarding of necessities is a common response to crises, especially in developed economies where there is normally an expectation of plentiful supply. This behaviour was observed internationally during the early stages of the Covid-19 pandemic. In the absence of actual scarcity, this behaviour can be described as 'panic buying' and can lead to temporary shortages. However, there have been few psychological studies of this phenomenon. Here we propose a psychological model of over-purchasing informed by animal foraging theory and make predictions about variables that predict over-purchasing by either exacerbating or mitigating the anticipation of future scarcity. These variables include additional scarcity cues (e.g. loss of income), distress (e.g. depression), psychological factors that draw attention to these cues (e.g. neuroticism) or to reassuring messages (eg. analytical reasoning) or which facilitate over-purchasing (e.g. income). We tested our model in parallel nationally representative internet surveys of the adult general population conducted in the United Kingdom (UK: N = 2025) and the Republic of Ireland (RoI: N = 1041) 52 and 31 days after the first confirmed cases of COVID-19 were detected in the UK and RoI, respectively. About three quarters of participants reported minimal over-purchasing. There was more over-purchasing in RoI vs UK and in urban vs rural areas. When over-purchasing occurred, in both countries it was observed across a wide range of product categories and was accounted for by a single latent factor. It was positively predicted by household income, the presence of children at home, psychological distress (depression, death anxiety), threat sensitivity (right wing authoritarianism) and mistrust of others (paranoia). Analytic reasoning ability had an inhibitory effect. Predictor variables accounted for 36% and 34% of the variance in over-purchasing in the UK and RoI respectively. With some

**Data Availability Statement:** All relevant data are within the paper and its Supporting Information files.

**Funding:** The initial stages of this project were supported by start-up funds from the University of Sheffield (Department of Psychology, the Sheffield Methods Institute and the Higher Education Innovation Fund via an Impact Acceleration grant administered by the university) and by the Faculty of Life and Health Sciences at Ulster University. The research was subsequently supported by the ESRC under grant number ES/V004379/1 and awarded to RPB, TKH, LL, JGM, MS, JM, OM, KB and LM. The funders had no role in study design, data collection and analysis, decision to publish, or preparation of the manuscript.

**Competing interests:** The authors have declared that no competing interests exist.

caveats, the data supported our model and points to strategies to mitigate over-purchasing in future crises.

## Introduction

During the early stages of the COVID-19 pandemic news outlets around the world reported what was widely described as "panic buying" of a wide range of household commodities, but especially toilet rolls [1], which led to temporary shortages in Australia, Italy, Japan, Singapore, Spain, the United Kingdom (UK) and the United States [2]. These observations have recently been confirmed in a study of bar code data in the UK [3] and in an analysis of credit card transactions in Australia [4]. Here we review the scant existing literature on consumer purchasing during crises, propose a psychological model of over-purchasing and then test it in representative samples from the populations of the United Kingdom (UK) and Republic of Ireland (RoI).

### Historical background and basic concepts

Although rarely recorded before the beginning of the twentieth century, excessive purchasing and hoarding of essentials, sometimes leading to scarcity of basic goods, has since been observed during many crises. This has especially been the case in populations living in comfortable circumstances in which there is ordinarily an expectation of unbroken access to essential commodities. It occurred, for example, during both world wars, the Cuban Missile Crisis, the 1979 oil crisis [5] and has also been observed during natural disasters such as earthquakes [6] and other pandemics [2]. When Spanish flu arrived in Britain immediately after the First World War, the rush to purchase quinine and other medications led to the threat of shortages [7], and during the 2003 SARS pandemic, over-purchasing in China and Hong Kong led to actual—albeit temporary—shortages of salt, rice, vinegar, vegetable oil, masks, and medicines [8].

It is important to note that human behaviour observed during a crisis, far from evidencing a panic-driven breakdown of the social order, is often adaptive [9]. Hoarding the necessities of life in anticipation of supply-side scarcity is a rational survival strategy. Indeed, in the US, the Federal Emergency Management Agency (FEMA) recommends that all households hold a stockpile of two weeks of non-perishable foodstuffs [10] and households are advised to take similar measures in earthquake-prone regions of Japan and New Zealand where non-compliance is seen as a social problem [11]. Over-purchasing, which we define as buying more than is necessary to sustain a household during routine life (see Table 1), primarily becomes problematic when it creates demand-side scarcity, stimulating further over-purchasing and potentially a vicious cycle of demand outstripping supply. In some circumstances, this can occur in

**Table 1. Definitions.**

| |
|---|
| 1. **Over-purchasing**: *buying more than is necessary to sustain a household during routine life*, as evidenced by increased purchasing compared to a previous period. Extreme over-purchasing may lead to demand-side scarcity, as stocks become depleted in retail outlets. |
| 2. **Panic buying**: *Over-purchasing in the absence of supply-side scarcity*. |
| 3. **Hoarding**: *The storing of essentials for use at a later time*. In both human and nonhuman animals, hoarding is an insurance policy against the event of future scarcity. |
| 4. **Scarcity cue**: *Any cue or information indicating the likelihood of future scarcity*. |

the absence of actual supply-side scarcity at the outset, in which case the term 'panic buying' seems appropriate.

For example, an economic study of the 2011 Tōhoku tsunami and earthquake in Japan, which used supermarket barcode data to compare household purchasing before and during the crisis, found considerable variation in the extent to which households over-purchased. Importantly, those households which excessively purchased did so across a wide range of commodities, implying a general tendency to over-purchase rather than the selective hoarding of specific necessities that were in danger of short supply [6].

Over-purchasing, panic buying and hoarding therefore appear to be social psychological phenomena that vary between individuals and therefore households. However, the mechanisms that lead to this behaviour have been subjected to very little psychological research. In a recent systematic review [12], 27 relevant publications were identified that were predominately within the business, management, and accounting literature. Many of these papers were non-empirical, concerned with responses to non-distaster related scarcity [13] focused on the performance of supply chains and retailers [14], or involved economic models [15], simulations [16], or surveys about likely behaviour in a crisis [11]. Four themes relating to potential psychological mechanisms were identified in the review: the perception of threat and scarcity, fear of the unknown, panic buying as a form of coping bahviour, and social influences such as the observed behaviour of others and lack of social trust.

In the context of the current coronavirus pandemic, some researchers have considered personality factors that may contribute to over-purchasing, such as the six-dimensions (honesty-humility, emotionality, extraversion, agreeableness, conscientiousness, and openness to experience) of the HEXACO model of personality [17]. In two internet surveys in the UK conducted in early March 2020 [18], the personality dimension of honesty-humility was modestly associated with negative responses to the question, "I have bought more food or supplies than usually" and with reduced intentions to over-purchase in future. These findings were interpreted as evidence that those individuals who refrained from over-purchasing were motivated to maximise societal outcomes, even if this required forgoing individual benefits. However, this finding was not replicated in a subsequent study with an internet convenience sample from 35 countries that focused specifically on toilet roll purchasing [19]. The authors of this study found that over-purchasing was predicted by the HEXACO dimension of conscientiousness–which was interpreted as evidence that more prudent individuals tend to stockpile–and also that an indirect association between emotionality and over-purchasing was mediated by anxiety about COVID-19.

We think that research on over-purchasing is likely to be more fruitful if based on a clear theoretical model. Here we outline a model that treats over-purchasing and panic buying as responses to the perceived threat of uncertain supplies, and report a preliminary test of our model using survey data collected from the UK and the Republic of Ireland (RoI) during the early stages of the 2020 COVID-19 pandemic.

## A consumer foraging model

Searching for food and other essential resources is a near-ubiquitous behaviour across the animal kingdom, including the human species [20–22] which has spent most of its evolutionary history in foraging economies [23]. In developed economies, the resources necessary for survival are typically obtained from shops and supermarkets. Consumers' choices about which supplier to visit typically obey a 'gravity model' (more often used to predict trading between nations [24] in which, other factors being equal, larger suppliers that are nearer to home are preferred [25, 26].

Research into animal patch foraging provides a framework for understanding how consumers evaluate the trade-off between exploiting known, local resources and travelling to distant locations where there is an unknown distribution of rewards [27]. In the simple context of supermarket purchases, this is the choice between buying goods from a local supermarket, which has an observable distribution of goods (e.g. canned goods, dried food), or sacrificing the cost in time and effort required to travel to a more distant supermarket where the abundance of these goods is unknown [28]. Decisions of this kind require the individual to track two parameters: the directly observable rewards that are available at the known patch, termed the *foreground rate*, and the rewards that can be expected from exploring the unknown patch, estimated from the organism's past encounters with similar patches in that environment, known as the *background rate* [29].

According to Charnov's marginal value theorem [20], in a stable environment the forager should explore a novel resource when the foreground rate falls below the background rate. This policy should maximise the ability to maintain energetic homeostasis, such that the organism does not expend more energy than it accumulates, nor acquires more energy than is necessary for survival [30]. However, this policy is likely to be ineffective or difficult to implement in an unstable environment in which the background rate is falling rapidly. Indeed, if the background rate is falling faster than the foreground rate, the marginal rate theorem mandates continued foraging in the home patch even though this strategy may ultimately lead to the patch becoming completely depleted and hence starvation. In the face of this kind of risk to survival, but before depletion occurs, accumulating energy for future use may be an effective insurance strategy that can be achieved in two ways: first, by hoarding supplies so that they can be retrieved later [31] and, second, by consuming more than is required to maintain energetic homeostasis [30]. Hence, priming human participants with cues that indicate future scarcity leads to increased consumption of high calorific food items [32]. Furthermore, in wealthy economies with easy access to these kinds of foods, distance to supermarkets where fresh produce is available [26] and food insecurity [33] are both associated with unhealthy diets and obesity.

A complication in this traditional account of foraging is that the background rate cannot be observed directly; it must therefore be inferred from whatever information is available. In a political crisis or natural disaster, judgments about the uncertainty of future supplies are likely to be informed by news reports of unfolding events [34, 35], which may be subject to rapid and unpredictable change, and also by the behaviour of other consumers. Hence, the depletion of stocks on supermarket shelves caused by people who have already engaged in panic buying may create a powerful *scarcity cue* that suggests that the availability of necessities in the future cannot be guaranteed. What begins with reports of shortages of particular products may therefore escalate into the panic buying of a wide range of goods. In many countries, these reported shortages began with toilet paper, possibly because they are large items that are most conspicuous when absent from the shelves [36] or because fear of the virus activates feelings of disgust [37] according to two speculative accounts. In an attempt to prevent further demand-side scarcity, politicians, emergency services, and supermarket managers may attempt to deliver reassuring messages (e.g. [38]) designed to persuade consumers that supplies will be rapidly replenished—that is, that the background rate is not falling.

The application of foraging theory to this context complements work that has examined the effects of manipulating product availability on consumers' behaviour. In fact, the manipulation of scarcity cuse is a common and effective marketing method used to increase the demand for products [39]. Foraging theory presents a lens through which to understand the efficacy of such marketing tactics; through presenting cues that the environment is poorer in resources (i.e. fewer items are available), the forager should infer a lower background rate and hence

forage the current patch more extensively [20]. Notably, it has been observed that it is the popularity of items, rather than their exclusivity that drives this scarcity effect, suggesting this relies on the behaviour of other consumers rather than retailers [40]. This is consistent with evidence that observing others engaging in panic purchasing positively associated with increased consumer's own over-purchasing during the COVID-19 pandemic [15]. Hence, the behaviour of other consumers can be an important scarcity cue indicating a fall in the background rate.

A further parameter that affects foraging behaviour in animals is the risk of predation. While viruses are parasites, rather than predators, the nature by which the virus spreads, i.e. through close contact with other human beings, should increase vigilance and subsequently lead to other people becoming associated with a threat to life [28]. In order reduce exposure to infection, individuals who perceive a high risk of infection should increase their foraging effort to hoard a greater amount of resources to maximise inter-foraging delays and minimise the frequency of encounters that put the individual at risk. Therefore, along with tracking scarcity cues, perceived risk of infection should predict over-purchasing behaviour.

## Demographic, situational and individual differences factors associated with over-purchasing, panic buying, and hoarding

Drawing on foraging theory, we have argued that over-purchasing, panic buying, and hoarding occur when individuals perceive that the background rate is falling rapidly, and that this perception is stimulated by scarcity cues in the form of news reports (balanced by reassuring messages) and also by directly observed cues such as shortages (see Fig 1). Using this framework, we identify three kinds of variables that may either exacerbate or mitigate against over-purchasing.

First, some demographic and situational factors may confer vulnerability to future scarcity. These factors can be considered to contribute to the overall salience of scarcity cues which, according to our foraging account, should motivate increases in purchasing behaviour. In this first type of variable, *exposure to the coronavirus* or *being close to someone else who has been exposed*, or pre-existing health conditions that confer vulnerability to self or someone close might be expected to signal reduced future access to shops, and therefore risk of scarcity. *Household size* might also be expected to have the same effect because a greater number of mouths to feed implies direr consequences if stocks cannot be replenished. Consistent with this account, household size predicted over-purchasing in the Tōhoku tsunami and earthquake [6]. Given that economic hardship [41] and food insecurity [42, 43] are a major source of psychological distress in parents, we predict that *households with children will show a greater propensity for over-purchasing*. The threat of *economic loss* associated with the pandemic, signalling reduced ability to secure supplies in the future, might also be expected to lead to a tendency to hoard while this is still possible (although see below). Food security in times of uncertainty might well depend on alliances with neighbours. As such, we also predict that the sense of belonging to a neighbourhood (which we have previously shown is protective against the stress associated with financial hardship [44]) and trusting relationships with neighbours will also mitigate against over-purchasing because these kinds of relationships will signal support from others in the event of depleted supplies. Conversely, paranoia (which we have previously shown is associated with harsh neighbourhoods in which trust is low [45]) should be associated with a greater tendency to over-purchase. Finally, we predict that *perceived risk of infection* should be associated over-purchasing, as this should increase effort to hoard resources in order to reduce the frequency of shopping trips and subsequently future risk of infection.

The salience of scarcity cues is likely to provoke negative emotional reactions. In the context of foraging theory, the stress response, provoked by aversive changes in the environment, should lead to greater exploitation of known patches [29]. Indeed, there is evidence that experimentally-induced

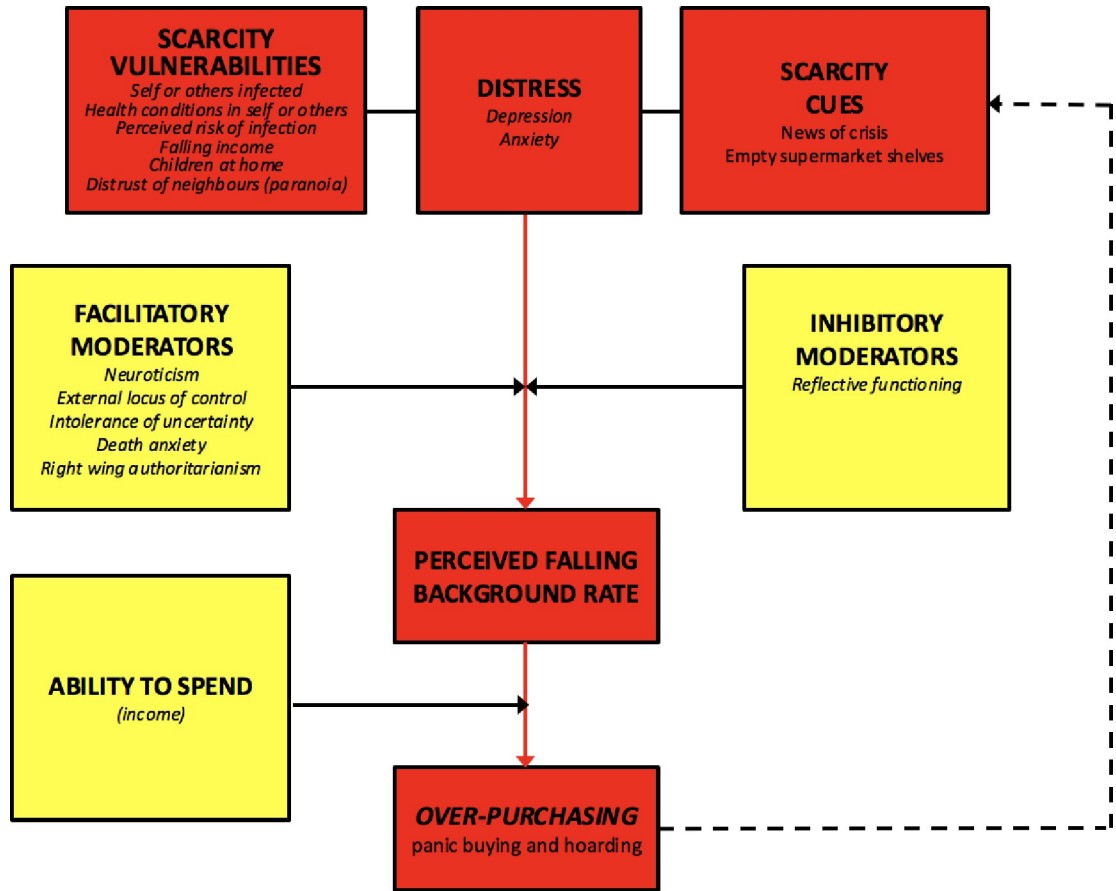

**Fig 1. Proposed model of factors influencing over-purchasing during a crisis.** Variables measured in the present study in italics.

stress, measured both physiologically and by subjective reports, is associated with the tendency to exploit a known resource for longer and engage in less frequent exploration [46]. Thus, indices of stress such as *depression*, *anxiety*, and *specific anxiety about the COVID-19 pandemic* (as reported by [19]) should be associated with greater over-purchasing.

A second type of variable that should influence over-purchasing is any individual difference factor that affects attention to scarcity cues or perceived risk of infection. These factors might include *neuroticism* [47], or traits which enhance perceptions of limited personal control over and uncertainty about the future, for example an *external locus of control* [48] and *intolerance of uncertainty* [49], or which indicate an enhanced perception of existential threat such as *death anxiety* [50]. Amongst political dispositions, *right wing authoritarianism* should also predict over-purchasing as this ideology is associated with sensitivity to both social [51] and existential threat [52]. Conversely, it is possible that some individual difference variables will predict a tendency *not* to panic buy. In particular, the *capacity to reflect* and think about reassuring messages [53, 54] when faced with signals of future uncertainty is likely to be an inhibitory factor.

Finally, it is important to consider variables that affect the *ability* to over-purchase given the perception that the background rate of supermarket goods is falling. Important amongst these is *household income*. In the Tōhoku tsunami and earthquake disaster, wealthier households were observed to over-purchase more [4]. Note that this analysis leads us to predict an apparently paradoxical effect: that over-purchasing will be greater in those households that are most wealthy but also in those in which household income is falling.

### Purpose of the present study and hypotheses

Here we report a test of the above account of over-purchasing using data collected in the early stages of the COVID-19 pandemic from two large population internet surveys conducted in different but economically comparable European countries–the UK and RoI–which, by design, administered parallel measures. Our hypotheses are:

First, that over-purchasing behaviour will be highly correlated across different groups of commodities; it should not be restricted to particular commodities. Hence, we predict that over-purchasing will be constitute a single latent trait that can be inferred from purchasing decisions about of a range of goods.

Second, that the following demographic and situational variables will be associated with over-purchasing: household income, loss of income due to the pandemic, having children at home, being infected by the virus, having a loved one who has been infected, distrust of neighbours, and paranoia. We also predict that over-purchasing will be associated with indices of psychological distress (depression, anxiety, specific anxiety about the COVID-19 virus) and also specific psychological variables likely to increase attention to scarcity cues: neuroticism, intolerance of uncertainty, and death anxiety.

Third, we predict that a greater sense of belonging to a neighbourhood, greater trust in neighbours, and greater capacity for analytic thinking/cognitive reflection will be negatively associated with over-purchasing.

## Materials and methods

### Participants and procedure

During the first phase of the COVID-19 Psychological Research Consortium (C19PRC) Study, nationally representative surveys of the general adult populations of the UK (N = 2,025) and RoI (N = 1,041) were collected between the 23rd and 28th of March and between the 31st of March and 5th of April, 2020, respectively. The beginning of these periods corresponded with, in the UK, 52 days after the first case of COVID-19 was confirmed and on the day that the UK Prime Minister announced a national lockdown at 8.30 in the evening and, in RoI, 31 days after the first confirmed case and two days after the Taoiseach (Irish Prime Minister) announced a national lockdown. Data collection was conducted via the Internet by the survey company Qualtrics using stratified quota sampling to ensure that the sample characteristics of sex, age, and household income in the UK, and sex, age and geographical location in RoI, matched the respective populations. Therefore, these data were collected within the first week of the strictest physical distancing measures being enacted in both countries.

By design, the two surveys used comparable measures whenever possible. Inclusion criteria for both samples were that participants be aged 18 years or older at the time of the survey and be able to complete the survey in English. Median completion times were 28.91 and 37.52 minutes for the UK and Irish surveys respectively. Ethical approval was granted by the University of Sheffield and Ulster University. All particiapnts provided written informed consent. The basic sociodemographic characteristics for both samples are reported by [55] and summarised in S1 Table. Quality checks were carried out on the data and are detailed elsewhere [50].

### Measures

Given the distinct socio-political contexts of the UK and Ireland, some variation existed in the measurement of the sociodemographic and political variables used in this study. All other variables were measured in an identical manner across the two samples.

**Demographic and household characteristics.** Self-reported gender and age were recorded, and age was also categorised into a 6-level variable for the regression analysis (18–24, 25–34, 35–44, 45–54, 55–64 and 65+ years).

Number of adults in household: Participants were asked "How many adults (18 years or above) live in your household (including yourself)?" and were provided with options ranging from '1' to '10 or more'.

Children: Participants were asked "How many children (below the age of 18) live in your household?" and were provided with options ranging from '0' to '10 or more'. The scores were categorised into 4 groups (0, 1, 2, and 3 or more children).

Income was categorised into quintiles using relevant economic data from the two countries (see [56]): in the UK survey participants were asked "Please choose from the following options to indicate your approximate gross (before tax is taken away) household income in 2019 (last year). Include income from partners and other family members living with you and all kinds of earnings including salaries and benefits" to choose one of 5 categories: "£0 - £300 per week (equals about £0 - £1290 per month or £0–15,490 per year)", "£301 - £490 per week (equals about £1,291 - £2,110 per month or £15,491 - £25,340 per year)", "£491 - £740 per week (equals about £2,111 - £3,230 per month or £25,341 - £38,740 per year)", "£741 - £1,111 per week (equals about £3,231 - £4,830 per month or £38,741 - £57,930 per year)", and "£1,112 or more per week (equals about £4,831 or more per month or £57,931 or more per year)". In the Irish survey participants were asked "Please choose from the following options to indicate your approximate gross (before tax is taken away) income in 2019 (last year)" and were provided with 10 categories: '0-€19,999', '€20,000-€29,999', '€30,000-€39,999', '€40,000-€49,999', '€50,000-€59,999', '€60,000-€69,999', '€70,000-€79,999', '€80,000-€89,999', '€90,000-€99,999', and '€100,000 or more'; these 10 categories were collapsed into 5 categories to align with the UK data.

Perceived household income changes during the COVID-19 pandemic: Respondents were provided with the following information, "Some people have lost income because of the coronavirus COVID-19 pandemic, for example because they have not been able to work as much or because business contracts have been cancelled or delayed. Please indicate whether your household has been affected in this way" and were provided three options: (1) My household has lost income because of the coronavirus COVID-19 pandemic, (2) My household has not lost income because of the coronavirus COVID-19 pandemic, and (3) I do not know whether my household has lost income because of the coronavirus COVID-19 pandemic. The last two categories were combined to produce a binary variable scored 1 = 'My household has lost income' and 0 = 'My household has not lost income/do not know'.

**Neighbourhood characteristics.** Three questions taken from the UK Community Living Survey [57] were asked of respondents to assess their level of belongingness and trust in relation to their neighbourhood. Neighbourhood belongingness was measured using the question "How strongly do you feel you belong to your immediate neighbourhood?" (scored on a 4-point scale from 1 'not at all' to 4 'very strongly'). Neighbourhood trust was measured by summing the responses to two questions, (1) "How comfortable would you be with asking a neighbour to keep a set of keys to your home for emergencies" and (2) "How comfortable would you be asking a neighbour to collect a few shopping essentials for you, if you were ill and at home on your own" (both scored on a 4-point scale ranging from 1 'very uncomfortable' to 4 'very comfortable').

**Over-purchasing.** Participants were asked, "Please indicate the degree to which you have increased your purchasing of the following items in recent weeks because of the COVID-19 pandemic?" with nine items for: tinned foods, water, sanitary products (e.g. hand sanitiser), toilet roll, dried food (e.g. pasta, rice), bread, pharmacy products (e.g. painkillers, cold/flu

products), batteries, and fuel (heating or car fuel). Response options were 1 'not at all'; 2 'very slightly'; 3 'moderately'; 4 'to a considerable degree'; and 5 'very considerably'.

**Health and COVID-19 related variables.**   *Health problems*. Participants were asked "Do you have diabetes, lung disease, or heart disease?" and the response options were 'Yes' (1) and 'No' (0). They were also asked "Do any of your immediate family have diabetes, lung disease, or heart disease?" and the response options were 'Yes' (1) and 'No' (0).

*Covid-19 status-self*. Participants were asked "Have you been infected by the coronavirus COVID-19?" and seven responses were provided, which were collapsed into a binary variable representing 'Perceived infection status—self'. Positive perceived infection status was based on the selection of either, 'I have the symptoms of the COVID-19 virus and think I may have been infected' or 'I have been infected by the COVID-19 virus and this has been confirmed by a test'. Negative perceived infection status was based on the selection of either, 'No. I have been tested for COVID-19 and the test was negative', 'No, I do not have any symptoms of COVID-19', 'I have a few symptoms of cold or flu but I do not think I am infected with the COVID-19 virus', 'I may have previously been infected by COVID-19 but this was not confirmed by a test and I have since recovered' or 'I was previously infected with COVID-19, this was confirmed by a test and I have now recovered'. Positive status (self) was coded '1' and negative status coded as '0'.

*Covid-19 status-other*. Participants were also asked "Has someone close to you (a family member or friend) been infected by the coronavirus COVID-19?" and four responses were provided. These were collapsed into a binary variable representing 'Perceived infection status–other'. Positive perceived infection status was based on the selection of either, 'Someone close to me has symptoms, and I suspect that person has been infected' or 'Someone who is close to me has had a COVID-19 virus infection confirmed by a doctor'. Negative perceived infection status was based on the selection of either, 'No' or 'Someone close to me has symptoms, but I am not sure if that person is infected'. Positive status (other) was coded '1' and negative status coded as '0'.

*Perceived risk of COVID-19 infection*. Participants were asked to estimate their personal percentage risk of being infected with the COVID-19 virus in the next month. A slider was presented with '0' and '100' at the left and right hand extremes respectively, and the labels 'No Risk', 'Moderate Risk' and 'Great Risk' were shown on the left, middle and right-hand part of the scale respectively. This produced a continuous score ranging from 0 to 100 with higher scores reflecting higher levels of perceived risk of being infected by COVID-19.

**Psychological distress.**   *Anxiety relating to COVID-19*. Respondents' degree of anxiety about the COVID-19 pandemic was assessed using a single visual slider scale, ranging from 0 'not at all anxious' on the left-hand side to 100 'extremely anxious' on the right-hand side.

*Depression*. Nine symptoms of depression were measured using the Patient Health Questionnaire-9 (PHQ-9; [58]). Participants indicated how often they had experienced each symptom over the previous two weeks using a 4-point Likert scale ranging from 0 'Not at all' to 3 'Nearly every day'. Possible scores ranged from 0 to 27, with higher scores indicative of higher levels of depression. The psychometric properties of the PHQ-9 scores have been widely documented, and the reliability of the scale among the current sample was excellent in the UK ($\alpha$ = .92) and Ireland ($\alpha$ = .91). Scores of 10 or more indicate depression of 'moderate' severity [59].

*Generalized anxiety*. Symptoms of generalized anxiety were measured using the Generalized Anxiety Disorder 7-item Scale (GAD-7; [60]). Participants indicated how often they had experienced each symptom over the previous two weeks on a four-point Likert scale (0 'Not at all' to 3 'Nearly every day'). Possible scores ranged from 0 to 21, with higher scores indicative of higher levels of anxiety and scores of 10 or more indicating anxiety of moderate severity. The GAD-7 has been shown to produce reliable and valid scores in community studies [61] and the reliability in the current sample was excellent in both the UK ($\alpha$ = .94) and Ireland ($\alpha$ = .94).

*Paranoia* was measured using the brief five-item version [62] of the persecution subscale of the Persecution and Deservedness Scale [63]. Example items include "People will almost certainly lie to me" and "You should only trust yourself". The scale scores had high reliability in the UK ($\alpha$ = .86) and Ireland ($\alpha$ = .83).

**Personality.** The Big-Five Inventory (BFI-10) [64] measures the 'big five' traits of openness to experience, conscientiousness, extraversion, agreeableness, and neuroticism widely found in studies of personality variation [65]. These traits correspond closely to five of the six traits assessed in the HEXACO model [17] assessed in two previous studies of over-purchasing [18, 19], the exception being honesty-humility. Each trait is measured by two items using a five-point Likert scale that ranges from 1 'strongly disagree' to 5 'strongly agree'. Higher scores reflect higher levels of each personality trait, and Rammstedt and John [64] reported good reliability and validity for the BFI-10 scale scores.

**Other psychological variables.** *Locus of control.* The Locus of Control Scale (LoC) [66] measures internal (e.g., 'My life is determined by my own actions') and external locus of control. The latter has two components, 'Chance' (e.g., 'To a great extent, my life is controlled by accidental happenings') and 'Powerful Others' (e.g., 'Getting what I want requires pleasing those people above me'). Each subscale was measured using three questions and a seven-point Likert scale that ranges from 1 'strongly disagree' to 7 'strongly agree'. Higher scores reflect higher levels of each construct. The scale scores had acceptable reliability in the UK (Chance, $\alpha$ = .70; Powerful Others, $\alpha$ = .85; Internality, $\alpha$ = .71) and Ireland (Chance, $\alpha$ = .63; Powerful Others, $\alpha$ = .78; Internality, $\alpha$ = .67).

*Intolerance of uncertainty.* Respondents' intolerance of uncertainty, which is thought to play a key role in the aetiology and maintenance of worry, was assessed using the 12-item Intolerance of Uncertainty Scale (IUS; [67]). The IUS was originally constructed to measure two factors, '*unexpected events are negative and should be avoided*' measured by items such as 'I always want to know what the future has in store for me', and '*uncertainty leads to the inability to act*' measured by items such as 'the smallest doubt can stop me from acting' [68]. Recent factor analytic research has shown that the IUS is best described by a bi-factor model, with a strong general factor being much more reliable than the specific factors, and hence unidimensional scoring is appropriate [69]. The scale had excellent reliability in the UK ($\alpha$ = .90) and RoI ($\alpha$ = .87).

*Death Anxiety Inventory.* Respondents' attitudes towards death were assessed using the 17-item Death Anxiety Inventory (DAI, [70]), which measures four death-related anxiety factors (labelled as death acceptance, externally generated death anxiety, death finality, and thoughts about death) with items such as 'I get upset when I am in a cemetery', 'The sight of a corpse deeply shocks me', 'I find it difficult to accept the idea that it all finishes with death' and 'I find it really difficult to accept that I have to die'. Responses were scored on a 5-point Likert scale ranging from 1 'totally disagree' to 5 'totally agree'. The scale had excellent reliability in the UK ($\alpha$ = .94) and Ireland ($\alpha$ = .92).

*Right wing authoritarianism.* The Very Short Authoritarianism Scale (VSA) [71] includes six items assessing agreement with statements such as: 'It's great that many young people today are prepared to defy authority' (reverse-scored) and 'What our country needs most is discipline, with everyone following our leaders in unity'. All items were scored on a five-point Likert scale ranging from 1 'strongly disagree' to 5 'strongly agree', with higher scores reflecting higher levels of authoritarianism. The internal reliability of the scale scores in the Irish sample was lower than desirable ($\alpha$ = .58) but somewhat stronger for the UK sample (a = .65).

*Analytical reasoning.* The Cognitive Reflection Task of Analytical Reasoning (CRT) [53] is a three-item measure of analytical reasoning where respondents are asked to solve logical problems designed to stimulate intuitively appealing but incorrect responses. The response format

was multiple choice with three foil answers (including the hinted incorrect answer), as recommended by [72].

## Data analysis plan

Means and standard deviations for the over-purchasing items were produced and the differences between Ireland and the UK were tested using independent samples t-tests. The magnitudes of these differences were assessed using Cohen's d estimates of effect size (d = 0.20 small effect, d = 0.50 medium effect, d = 0.80 a large effect). The main analysis was then conducted in three linked phases.

First, an exploratory factor analysis (EFA) was conducted on the over-purchasing data from Ireland and the UK separately. The number of factors to retain was based on the relative size of the sample eigenvalues and the 95 percentile eigenvalues from a parallel analysis with 500 replications. Models with 1 through to 3 factors were fitted, the parameters were estimated using robust maximum likelihood, and an oblique rotation (Geomin) was used for solutions with more than one factor.

The EFA of the over-purchasing items for both the UK and RoI strongly supported a unidimensional solution. The sample eigenvalues for the first three factors showed a very large first factor, and larger than the 95th percentile eigenvalues (in parentheses). In the UK: factor 1 = 6.335 (1.130); factor 2 = 0.763 (1.090), factor 3 = 0.384 (1.060); in Ireland, sample eigenvalues were as follows: factor 1 = 5.351 (1.185), factor 2 = 0.815 (1.127), factor 3 = 0.594 (1.084). These findings are strong evidence of a single latent trait of over-purchasing.

Second, a multi-group model was specified for the combined Irish and UK data, with the same factor structure (configural invariance) imposed and the factor loadings constrained to be equal across countries (metric invariance). The predictor variables were added, and the country-specific regression coefficients were estimated.

In the third phase, the between-group regression coefficients were tested for differences using the Wald test: if the coefficients were not significantly different, the paths were constrained to be equal; if they were significantly different, the coefficients were allowed to vary across groups. All of these analyses were conducted using latent variable modelling in Mplus 8.1 [73].

## Results

The distribution of total over-purchasing scores for the UK and RoI, are shown in Fig 2. Mean scores for the individual over-purchasing items together with tests for differences, are reported in Table 2. These data indicate that the highest rates of over-purchasing were reported by only a minority of both populations, with scores skewed towards the right end of the distribution. For example, in the UK sample, 23.6% recorded a score of 1 for all 9 items, indicating no over-purchasing at all, and this was the most common response; the corresponding figure for RoI was 9.1%. 70.3% of the UK sample had mean item scores of less than 2 ("very slightly), indicating very modest over-purchasing behaviour and the corresponding figure for RoI was 53.3%.

Scores were higher in RoI than the UK for all items, and these differences were significant with the single exception of tinned food. In absolute terms, however, these differences were generally not large (the exception being sanitary products) and the fact that they were statistically significant should be interpreted in the context of the large sample sizes. We also compared total purchasing scores for living location (city, n = 753; suburb, n = 760; town n = 918; and rural area, n = 635). There was a significant main effect, F(3, 3062) = 34.73, p < .001, although the effect size was small ($h^2$ = .03). The mean total purchasing score was highest for people who described themselves as living in a city (mean = 19.28, SD = 9.25) and was

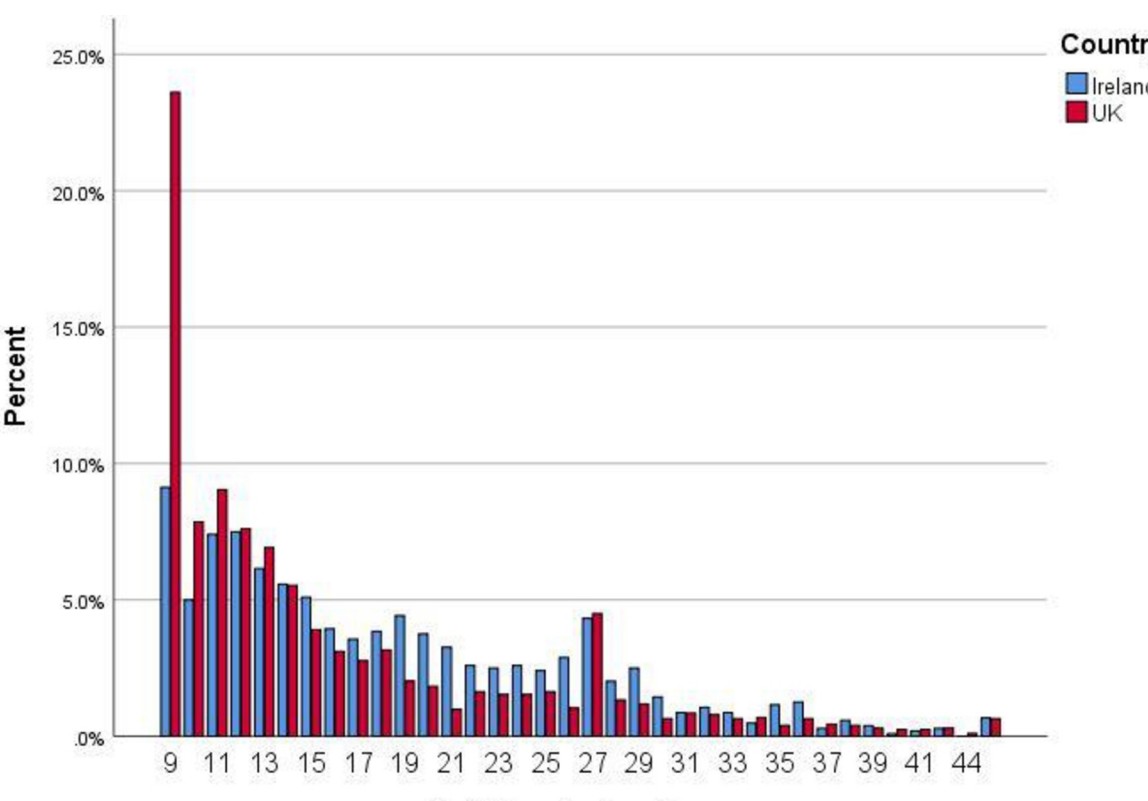

**Fig 2. Distribution of over-purchasing in the UK and ROI during the early stages of the COVID-19 pandemic.** Total scores on a 9-item scale with a minimum score of 9 and a maximum score of 45. Note: individual items were scored 1 'not at all'; 2 'very slightly'; 3 'moderately'; 4 'to a considerable degree'; and 5 'very considerably'. Hence, mean item scores of < 2 and mean total scores of < 18 imply little or no over-purchasing.

significantly higher (Scheffe, P < .001) than for those who said they lived in a suburb (mean = 16.14, SD = 7.71), town (mean = 16.13, SD = 7.56) or rural area (mean = 15.43, SD = 6.97). No other pairwise differences were significant (all p > .05).

Zero-order correlations between factor scores, country-specific standardised regression coefficients and standardised regression coefficients, and Wald tests for the multi-group

**Table 2. Descriptive statistics for panic buying items from Ireland and UK.** Mean item scores of < 2 imply little or no over-purchasing.

| | Ireland N = 1041 | UK N = 2025 | t | p | Cohen's d |
|---|---|---|---|---|---|
| Tinned food | 2.08 (1.10) | 2.01 (1.10) | 1.652 | .099 | .061 |
| Water | 1.90 (1.25) | 1.54 (1.03) | 8.586 | .000 | .328 |
| Sanitary products (hand sanitiser) | 2.68 (1.28) | 1.91 (1.13) | 17.036 | .000 | .650 |
| Toilet roll | 2.17 (1.18) | 1.96 (1.12) | 4.928 | .000 | .188 |
| Dried foods (e.g. pasta. rice) | 2.31 (1.17) | 2.00 (1.11) | 7.185 | .000 | .274 |
| Bread | 2.00 (1.15) | 1.79 (1.05) | 5.222 | .000 | .199 |
| Pharmacy products (e.g. painkillers, cold/flu products) | 2.02 (1.12) | 1.76 (1.03) | 6.608 | .000 | .252 |
| Batteries | 1.57 (0.98) | 1.42 (0.92) | 4.183 | .000 | .160 |
| Fuel (heating or car fuel) | 1.76 (1.07) | 1.49 (0.97) | 7.152 | .000 | .273 |

Note: df = 3064 for all t-tests

**Table 3. Correlations and standardised regression coefficients for model of predictors of hoarding latent variable.**

| | Ireland | UK | Ireland | UK | Wald Δ Ire-UK | Multi-group estimates (Ireland/UK) |
|---|---|---|---|---|---|---|
| | r | r | β | β | | |
| Age | -.282** | -.274*** | -.177*** | -.141*** | 0.660 | -.150*** |
| Gender (male) | .010 | .036 | .037 | .086*** | 2.031 | .074*** |
| Number adults | .097*** | .089*** | .041 | .031 | 0.020 | .027 |
| Number children | .167*** | .239*** | .076* | .065** | 0.002 | .069*** |
| Income | .046 | .006 | .066* | .052* | 0.685 | .059** |
| Lost income | .130** | .120*** | .055 | .024 | 0.648 | .033* |
| Neighbourhood belonging | -.005 | .121*** | .077* | .147*** | 3.708 | .127*** |
| Neighbourhood Trust | -.072* | -.039 | -.027 | -.034 | 0.045 | -.033 |
| Intolerance of Uncertainty | .244*** | .234*** | .021 | .007 | 0.095 | .014 |
| LOC: Chance | .187*** | .235*** | -.023 | -.040 | 0.154 | -.033 |
| LOC: Powerful Others | .114*** | .343*** | -.008 | .126*** | 4.136* | .005/.116*** |
| LOC: Internal | .111*** | -.041 | .011 | .034 | 0.295 | .023 |
| Paranoia | .370*** | .371*** | .194*** | .142*** | 1.132 | .162*** |
| Depression (>10) | .200*** | .282*** | .030 | .087** | 1.330 | .068** |
| Anxiety (>10) | .200*** | .244*** | -.002 | .024 | 0.262 | .017 |
| Extraversion | .001 | .083*** | .050 | .080*** | 0.720 | .070*** |
| Agreeableness | -.086*** | -.128*** | .048 | -.012 | 2.810 | .007 |
| Conscientiousness | -.149*** | -.196*** | -.062* | -.103*** | 1.204 | -.089*** |
| Neuroticism | .157*** | .108*** | -.010 | -.120*** | 6.692** | -.007/-.130*** |
| Openness | -.117*** | -.033 | -.076** | -.015 | 4.038* | -.082**/-.014 |
| Right Wing Authoritarianism | .136*** | .056* | .101*** | .044* | 3.200 | .062*** |
| Death Anxiety | .380*** | .407*** | .192*** | .210*** | 0.020 | .210*** |
| Cognitive Reflection Task | -.202*** | -.186*** | -.108*** | -.089*** | 0.561 | -.093*** |
| COVID-19 Anxiety | .140*** | .122*** | .029 | .009 | 0.252 | .014 |
| Health problems—self | .050 | .038 | .024 | .008 | 0.191 | .011 |
| Health problems—other | .124*** | .025 | .069* | .006 | 2.751 | .026 |
| Perceived infection status—self | .026 | .101*** | -.004 | .044* | 1.755 | 0.032 |
| Perceived infection status—other | .080** | .085*** | .018 | .025 | 0.054 | 0.022 |
| Personal Risk | .260** | .252*** | .120*** | .098*** | 0.275 | 0.107*** |
| R-square | | | .341*** | .364*** | | .320***/.363*** |
| Adjusted R-square | | | .335 | .358 | | .314/.357 |

model as shown in Table 3. A complete correlation table showing relationships between all predictor variables is available in S2 Table. For the initial multi-group model, factor loadings were specified as invariant across the groups and the standardised factor loadings ranged from .645 to .789 and all were statistically significant ($p < .001$). The over-purchasing latent variable was regressed on the predictor variables, and the group-specific regression coefficients were estimated. As explained in the analysis plan, where significant differences existed in these coefficients for the two countries, coefficients are reported separately.

Of the 23 variables specified in our model, all but six of the zero-order correlations were significant and in the direction expected in one or (in the majority) both countries; only one variable had a significantly opposite association to that predicted (internal locus of control was positively associated with over-purchasing in the Irish sample).

Many but not all of the final multi-group estimates also support our hypotheses, accounting for 36% (UK) and 32% (RoI) of the variance in over-purchasing. In this model, only

neighbourhood belonging (in both countries) and neuroticism (in the UK) behaved contrary to expectation.

With respect to demographic variables, over-purchasing was associated with being younger, female, having children in the home, and having higher income but also with having lost income because of the pandemic. Contrary to expectation, health and infection status of self or others close to the self were not associated with over-purchasing, although a global measure of perceived risk of infection was. Although distrust of neighbours was not significant in the final model, the more general measure of paranoia was.

Of the psychological distress variables, only depression and death anxiety were significant in the multi-group model, although generalized anxiety and specific anxiety about the corona-virus were significant predictors when zero-order correlations were considered. Most likely this was a suppression effect caused by including multiple variables that included an anxiety component. This effect may also explain why neuroticism was negatively associated with over-purchasing in the UK in the multi-group model, despite being positively correlated with over-purchasing in both countries. There were also small effects for two of the other personality dimensions–extraversion and low conscientiousness.

Right wing authoritarianism was associated with over-purchasing, as expected, but the other psychological variables (the locus of control dimensions, intolerance of uncertainty) mostly failed to contribute to the multi-group model, despite being correlated with over-purchasing in the manner expected. Finally, as predicted, capacity for analytical reasoning (the CRT) was negatively associated with over-purchasing.

## Discussion

We have proposed a psychological model of over-purchasing and panic buying, which was tested using large, nationally representative datasets from the UK and RoI collected in the early phase of the COVID-19 pandemic. To our knowledge, this is the largest and most comprehensive study of its kind, and the findings have the potential to inform policy in future national and international emergencies. The predictor variables included in our models were selected in light of the theoretical account of over-purchasing that we outlined in our introduction. However many were also consistent with previous speculation about the psychological factors involved in this phenomenon, such as personality and anxiety towards COVID-19 [12, 74, 75], but some had never before been considered in this context, for example death anxiety and analytical reasoning.

Substantial over-purchasing was reported by a minority of the participants across the two samples; about three quarters reported no or very little over-purchasing at all. This is consistent with evidence that stockouts during the early stages of the pandemic were driven by large numbers of consumers increasing their purchasing by modest amounts rather than a minority engaging in extreme levels of panic buying [3]. Nonetheless, we found strong evidence that over-purchasing is a coherent trait such that individuals who reported over-purchasing some items generally reported over-purchasing many items; a finding which is consistent with one of the few economic studies of panic buying [6]. Interestingly, toilet rolls were not subject to over-purchasing more than other commodities which suggests that over-purchasing is not related to disgust, as previously suggested [1]. The implication is that there must be one or more psychological processes that lead to a general tendency to purchase more than usual in times of crisis, which is consistent with our account that sees acquiring additional resources as a generalised and biologically adaptive response that is activated when the background rate–the availability of resources at reachable patches/supermarkets–is perceived to be falling rapidly.

We observed greater over-purchasing in RoI compared to the UK despite the fact that the two surveys were conducted at comparable stages in the pandemic, although, in absolute terms, these differences were generally not large (sanitary products being an exception). One possible explanation is that the government of Ireland took decisive action to contain the pandemic sooner than the UK government and, therefore, scarcity cues were more evident in that country when the surveys were conducted. In support of this, recent research has demonstrated that the timing of Government interventions are associated with panic buying, with earlier interventions coinciding with heightened rates of purchasing behaviours [76]. Commenting on differences between the nations, Irish historian Elaine Doyle was quoted in the UK's Guardian newspaper [77]:

> "While Boris [Johnson, the British Prime Minister] was telling the British people to wash their hands, our Taoiseach was closing the schools. While Cheltenham [a British horseracing festival] was going ahead, and over 250,000 people were gathering in what would have been a massive super-spreader event, Ireland had cancelled St Patrick's Day,"

We also found that over-purchasing was more likely to be reported by people living in urban areas compared to those living suburbs, towns or rural areas. Whilst we did not make a prediction about this, one possible explanation is that scarcity cues (including the visible behaviour of other consumers) are more available in urban environments.

Based on our model, we made predictions about demographic, situational, and psychological variables that would either facilitate or inhibit over-purchasing by influencing either scarcity cues, perceived risk of infection, perceptions of the background rate (future scarcity), or the ability to over-purchase. Many but not all of our findings were consistent with our predictions. Consistent with previous economic research and foraging theory [6], we found that over-purchasing was associated with household income (presumably reflecting the opportunity afforded by having the necessary financial resources). However, it was also associated with loss of income due to the pandemic, which might at first seem paradoxical, but is clearly consistent with a psychological mechanism by which hoarding is provoked by fear of future scarcity. The number of adults in the household did not predict over-purchasing, but the number of children did, which we predicted on the basis that food-insecurity is a major source of anxiety for parents [42, 43].

Against expectation, health-related variables (whether or not individuals, or those close to them, had either been infected or were vulnerable because of health difficulties) did not influence over-purchasing much, although there was a strong association between over-purchasing and the perceived risk of being infected in the future. It is possible that the failure of the infection-status variables to predict was because these referred to whether or not individuals or those close to them had been infected in the past and, therefore, could not signal the likelihood of future scarcity whereas risk of infection, which referred to a possible future event, could. This was consistent with our foraging framework, as perceived risk of infection should lead to increased hoarding to reduce the number of shopping trips required and subsequently the risk to oneself [28].

Of our measures of psychological distress, depression most clearly predicted over-purchasing, as expected. Previous research has reported a modest effect for specific anxiety about the coronavirus [19] but, although zero-order correlations were found between over-purchasing and both our generalised anxiety measure and also our measure of anxiety about COVID-19, these were not evident in the final multi-group model. There was, however, a large effect for death anxiety.

The only two previous psychological studies of over-purchasing during the present pandemic that we are aware of focused on personality and reported conflicting results. One found

that over-purchasing was inhibited by honesty-humility [17] (which we did not measure in this study, which began data collection before this finding was published), and hence prosociality, and the other did not find this effect [19]. These authors [19] also found that high conscientiousness was associated with over-purchasing whereas we found the opposite effect (which might be interpreted as conscientious people being more able to think through the long-term implications of their actions (i.e., that hoarding leads to scarcity), an account that would be consistent with our findings from the CRT), or that they were simply more complaint with advice not to purchase beyond their needs. Emotionality indirectly affected over-purchasing [19] whereas, in our study, neuroticism was positively associated with over-purchasing when zero-order correlations were considered, but negatively associated in the United Kingdom in the multi-group model (which we interpreted as a suppression effect). A reasonable conclusion, given these conflicting findings, is that there is, overall, not much evidence that the standard dimensions of personality play a large role in over-purchasing.

As predicted, over-purchasing was associated with right-wing authoritarianism, which manifests itself as increased sensitivity to threat and associated feelings of uncertainty [78, 79]. We have previously demonstrated that authoritarians are highly sensitive to the existential threat created by the COVID-19 pandemic; anxious authoritarians were found to be most likely to express nationalistic and anti-immigrant attitudes [52]. Hence, it appears that the stockpiling of essential items may serve as one way that authoritarians can bring order and security back into their lives.

One of our two predictions about inhibitors of over-purchasing was supported: in both countries over-purchasing was negatively associated with scores on an established measure of analytical reasoning [53]. Performance on this test and other measures of analytic reasoning have previously been associated with less willingness to believe in fake news or irrational beliefs such as conspiracy theories [80–82]. Our findings for trust, particularly in terms of relationships with neighbours, however, were at best mixed in relation to our hypotheses. On the one hand, as expected, paranoia—a measure of extreme distrust about the intentions of others—strongly predicted over-purchasing. This makes sense in the context of demand-side shortages when likelihood of scarcity is increased if other people act without regards to one's own interests. On the other hand, specific trust of neighbours did not predict over-purchasing and, unexpectedly, a sense of belonging to a neighbourhood was positively associated with over-purchasing. One possibility is that people in highly cohesive neighbourhoods tend to shop in the same supermarkets and talk amongst themselves about the shortages they observe (that is, they provide each other with scarcity cues), which is consistent with the previously noted observation that over-purchasing occurred in areas of high population density. This interpretation is also supported by work demonstrating that observing neighbours engage in panic buying increases consumers' own purchasing behaviours [15]. In future studies it will be interesting to investigate these variables in the context of supply-side shortages (for example, in the event of trade disruptions due to international disputes).

There are a number of strengths and limitations of this study. The major strength is that it is the most comprehensive study on this topic to date, which used high quality data collected from large and representative samples in two countries, collected early in the pandemic. We proposed and tested a psychological model of over-purchasing, which provides a framework for understanding how demographic, situational and psychological factors might exacerbate or mitigate over-purchasing behaviour. A limitation of the study is that one of the key concepts in the model–the perceived fall in the background rate (expectations about future scarcity)—was not directly measured, so the test was indirect. In future studies it will be important to include a variable in which people are asked to estimate the future likelihood of shortages, and also to ask people about their actual observations of news reports and empty supermarket

shelves. Another weakness of the study mirrors one of its strengths–the large number of variables that were available to us. In future studies, it will be useful to refine our model to explicitly address the inter-relationships between the variables and the possibility that some (for example, the anxiety measures) are tapping a common latent process. Also, methods to avoid potential bias from the use of common methods (self-report) will be considered in studies. Finally, it would be important for future studies to test whether these findings extend to countries outside of Europe, including the global South and in populations that are not western, educated, industrialised, rich and democratic (WEIRD; [83]).

Our model has implications for future crises. At a national level, governments, and at a local level, supermarket managers, should anticipate and seek to control over-purchasing by prohibiting bulk-buying, should manage scarcity cues in a way that reduces their salience (for example, by giving careful thought to the way that shelves are stocked), and should facilitate analytical reasoning; in their customers (for example, by providing detailed information about when dwindling stocks will be replaced). Our findings point to profiles of individuals who might be particularly vulnerable to over-purchasing, and who could either be reassured by appropriate policies–for example, by providing parents with young children special times when they can shop–or skilfully targeted messaging. Forward planning based on our model in preparation for future crises may lead to less panic-buying and less demand-side scarcity, and thereby may reduce the number of problems faced by governments during challenging times.

## Supporting information

**S1 Table. Sociodemographic characteristics of the Irish and UK samples.**
(DOCX)

**S2 Table. Correlation matrix of predictor variables.**
(DOCX)

**S1 Data.**
(SAV)

## Author Contributions

**Conceptualization:** Richard P. Bentall, Alex Lloyd, Kate Bennett, Ryan McKay, Liam Mason, Jamie Murphy, Orla McBride, Todd K. Hartman, Jilly Gibson-Miller, Liat Levita, Anton P. Martinez, Thomas V. A. Stocks, Sarah Butter, Frédérique Vallières, Philip Hyland, Thanos Karatzias, Mark Shevlin.

**Formal analysis:** Mark Shevlin.

**Investigation:** Richard P. Bentall, Alex Lloyd, Kate Bennett, Ryan McKay, Liam Mason, Jamie Murphy, Orla McBride, Todd K. Hartman, Jilly Gibson-Miller, Liat Levita, Anton P. Martinez, Thomas V. A. Stocks, Frédérique Vallières, Philip Hyland, Thanos Karatzias, Mark Shevlin.

**Writing – original draft:** Richard P. Bentall, Alex Lloyd, Kate Bennett.

**Writing – review & editing:** Richard P. Bentall, Alex Lloyd, Kate Bennett, Ryan McKay, Liam Mason, Jamie Murphy, Orla McBride, Todd K. Hartman, Jilly Gibson-Miller, Liat Levita, Anton P. Martinez, Thomas V. A. Stocks, Sarah Butter, Frédérique Vallières, Philip Hyland, Thanos Karatzias, Mark Shevlin.

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
