## [Decision Letter · Decision Letter 0]

19 Oct 2020

PONE-D-20-25983

Pandemic buying: Testing a psychological model of over-purchasing and panic buying  using data from the United Kingdom and the Republic of Ireland during the early phase of the COVID-19 pandemic

PLOS ONE

Dear Dr. Bentall,

Thank you for submitting your manuscript to PLOS ONE. After careful consideration, we feel that it has merit but does not fully meet PLOS ONE’s publication criteria as it currently stands. Therefore, we invite you to submit a revised version of the manuscript that addresses the points raised during the review process.

We look forward to receiving your revised manuscript.

Kind regards,

Frantisek Sudzina

Academic Editor

PLOS ONE

Reviewers' comments:

Reviewer's Responses to Questions

**Comments to the Author**

1. Is the manuscript technically sound, and do the data support the conclusions?

Reviewer #1: Yes

Reviewer #2: Partly

Reviewer #3: Yes

2. Has the statistical analysis been performed appropriately and rigorously? 

Reviewer #1: Yes

Reviewer #2: I Don't Know

Reviewer #3: Yes

3. Have the authors made all data underlying the findings in their manuscript fully available?

Reviewer #1: Yes

Reviewer #2: Yes

Reviewer #3: Yes

4. Is the manuscript presented in an intelligible fashion and written in standard English?

Reviewer #1: Yes

Reviewer #2: Yes

Reviewer #3: Yes

5. Review Comments to the Author

Reviewer #1: I had the pleasure of reading this interesting paper. As authors mentioned that “this is the largest and most comprehensive study of its kind”, I tend to agree that it is a large study of pandemic buying, both from the sample size point of view and also with regard to aspects considered, produced results are impressive. In my opinion, the paper did contribute to understanding why pandemic buying happens. However, there are points which in my opinion could further improve the paper:

• Some parts have contradicting claims. In the result part, authors claimed that “Table 2 indicate that over-purchasing was reported by only a minority of both populations” (line 476). However, at the beginning of the paper, from line 56 to line 59, the author used one reference (1) to indicate that “panic buying” especially short of “toilet rolls” were happening, including in the UK. Authors provided explanation in the discussion why one nation is more caution than another, but did not clearly mention in the background whether panic buying was serious in UK or ROI. There is need to provide more evidence or report to agree or argue that the pandemic buying existed in UK along explanation to explain why the survey opposed the evidence.

• In the discussion part. When authors mentioned that “many were consistent with previous speculation about the psychological factors involved in this phenomenon”. It is needed to clarify what was consistent with which specific previous analysis? Currently, there is only have one reference for this statement.

• In discussion: “The country was particularly badly affected by the 2008 financial crisis but, probably more importantly, historical education in RoI places emphasis on the Great Famine of 1845-1849, which led to the death through starvation or disease of about one million people and the emigration of another million, resulting in a reduction in the Irish population of about 25% over just five years (69). As a consequence, scarcity cues may be more salient for people in Ireland compared to people in the UK”. This sound as a bold and subjective conclusion. I belive Britain also suffered from the pandemic in 1918/19, estimating 228,000 deaths. To conclude that RoI is more salient there is need to provide evidence and elaborate it. This question links back to my previous question, why the survey showed a minority of panic buying when it happened. Would this be related to “Personality” that authors mentioned? Is it possible that people are reluctant to admit they did “pandemic buying”?

• There is need to reference some recent relevant work:

o Prentice, C., Chen, J., & Stantic, B. (2020). Timed intervention in COVID-19 and panic buying. Journal of Retailing and Consumer Services, 57, 102203.

• Zheng, Rui, Shou, Biying, Yang, Jun, 2020. Supply disruption management under consumer panic buying and social learning effects. Omega 102238

• ANZ Research, 2020. Panic buying or the essentials intensifies. Retrieved from. https ://www.savings.com.au/credit-cards/new-data-shows-massive-increases-and-d ecreases-in-consumer-spending-due-to-covid-19.

• Why particularly to survey UK and RoI? What is the similarity/difference in these two specific countries that worth be compared? Would be better that authors compare the UK with a place that is not in Europe? This needs to be addressed or mentioned as limitation.

• Another aspect is with distribution of data, while it is mentioned that different age, sex, etc, has been considered, it would be also very usefull to show how panic buying was perceived in urban and regional areas. Other work presented as a one of the important factors.

• There should be a reference not a link, line 75.

Reviewer #2: The manuscript describes how several psychological factors are related consumers’ reported over-purchasing and panic buying in the early phase of COVID-19 pandemic in UK and Republic of Ireland. The study is based on representative surveys carried out at the end of March/ beginning of April with good sample sizes. The study is highly topical and addresses how individual responses vary in an exceptional situation using a wide battery of psychological measures as possible explanations.

Overall, the manuscript is clearly structured, but there are several long sentences that are difficult to follow as they contain several points that would benefit from separating them in two sentences (or other re-writing; e.g. lines 59-63, 95-100, 105-111, 127-131, 143-147, 155-159, 167-171, 201-206).

Although methods are described to some detail, especially the measures, the reporting raises some questions related to data quality that reader would like to know. What explains the large difference in completion times between the countries? What was done to check the data quality? Was the final data representative of the two countries? Did you have to exclude some responses due to too short response time?

One of the strengths and weaknesses of the study are the many variables used to describe respondents’ living conditions, relation to COVID-19 pandemic, and psychological characteristics. However, there is very little information how these many independent variables relate to each other and the authors should provide a correlation matrix on this. One would expect distress, anxiety and depression symptoms to correlate positively with each other and with neuroticism, and further with the more domain specific measures, such as COVID 19 anxiety. The authors indicate that it would be interesting to study these relationships in the future, but that raises the question: why not do it with these data rather than collect new data or is the aim to use these same data in further analysis?

Currently the model contains variables that link very differently to over-purchasing. Why not run them in a hierarchical model to see how the different types of variables contribute to the overall explanation? Some are related to perceived infection status: although this may be a (very good) cause to over-purchase, it is an odd one to include in this kind of model as it does not relate to perception of oneself or living conditions in general. It relates to a short period of time which may have a very different impact whether the disease is on at the moment or has already been conquered. It would be important to know, how many people actually reported that they have had or thought to have had the disease: in the beginning of pandemic these cases are likely to be very low.

The over-purchasing variable seems to have very skewed distribution. How was this dealt with in the regression analysis? Would it had been interesting to explain who are those 20% reporting to over-purchase?

Conscientiousness being negatively related to over-purchasing may be linked to ability to think about the long-term consequences or it may simply be that respondents who were more conscientious were also that in following the official/governmental recommendations against hoarding.

Overall, the discussion does not go very deeply into the relative importance of the different variables in the strength of associations. In the limitations, the authors hint that looking at the relationships between variables would be of interest: perhaps the more general psychological variables mediate the relationships between COVID-19 based worries and over-purchasing. This would suggest that the current analysis of variables could go much further.

Minor points:

Lines 197-199: Why there is a direct link with children in the household and economic hardship and food security? Would this depend on the economic status of the household?

Line 342: Heath problems should read Health problems

Lines 421-429: reliability of the IUS should be reported similar to other scales

Lines 494-495: This sentence should be included in Table 2 to make it self-explanatory

Lines 507-512: This would be better placed in data analysis part as a conclusion to explaining the factor analyses

Lines 542-543: referring to earlier findings: move to discussion

Table 3: are the reported R2 values referring to adjusted values; there is an additional column in the Table on page 27. Make the Table self-explanatory: write out CRT

Discussion, page 29: referring to the Great Famine in Ireland in mid 1800 as a possible explanation for country differences sounds a bit far-fetched. How can one align the situation in Ireland now vs then? Why not other events or World War 2 experiences? I suggest to omit this.

Reviewer #3: While the authors are to be commended on introducing the biological theory, I find its application not convincing: They write that “In the simple context of supermarketpurchases, this isthe choice between buying goods fromalocal supermarket, which has an observable distribution of goods (e.g. canned goods, dried food), or sacrificing the cost in time and effort requiredto travel to a more distantsupermarket where the distribution of these goods is unknown.” I do not think this is true because a consumer will likely find the same supermarkets even when shopping elsewhere, being equipped with the very same products. It is quite a stretch to propose that this is ‘unknown’ to the consumer. As this is a key theoretical framework, I find it problematic that the argument is not convincingly applied to consumer realities.

Further, I am not sure whether the theoretical lens is sufficient to motivate the hypotheses and included variables (also see below). I would also suggest that the authors broaden their theoretical discussions and include the literature on consumer behavavior on scarcity effects more (there is a lot, e.g.: Hamilton, R., Thompson, D., Bone, S., Chaplin, L. N., Griskevicius, V., Goldsmith, K., ... & Piff, P. (2019). The effects of scarcity on consumer decision journeys. Journal of the Academy of Marketing Science, 47(3), 532-550.)

A weakness of the current paper is that the interesting front-end (using foraging theory) has not logical links with the hypotheses and empirical design of the study. Such a gap is problematic. It is not clear how the introduced and applied theory helps to develop the hypotheses. Clearly, the theoretical foundation is a biological one, but the hypotheses refer to household income or personality dimensions. I am not saying that this can not be derived from the theory but the authors just make to little effort to link these parts. This needs significant revision.

Also, the authors need to develop each individual hypothesis in more detail. They include various interesting variables, I am not doubting the suitability. However, the selection seems abitrary.

Related to this, I really miss a narrative of this research: It appears that a lot of variables have been tested without sufficient theoretical reasoning and a guiding theoretical framework. The paper would benefit significantly from a sharper profile.

As for the method, have the authors controlled for common method bias? And other potentially biasing response behavior?

Also, can the authors reason their choice of method, e.g., why a SEM has not been used.

Overall, the authors draw on a creative theoretical basis but there is a logic gap between this section and the empirical design. This is problematic but may be alleviated through a thorough revision. Good luck.

6. PLOS authors have the option to publish the peer review history of their article (what does this mean?). If published, this will include your full peer review and any attached files.

Reviewer #1: **Yes: **Professor Bela Stantic

Reviewer #2: No

Reviewer #3: No

---

## [Author Response · Author response to Decision Letter 0]

15 Jan 2021

Dear Dr. Sudinza

Many thanks for the opportunity to resubmit our manuscript entitled “Pandemic buying: Testing a psychological model of over-purchasing and panic buying using data from the United Kingdom and the Republic of Ireland during the early phase of the COVID-19 pandemic” (your ref: PONE-D-20-25983). We would like to thank the reviewers for their insightful and constructive comments, which have been extremely helpful in guiding the revisions we have undertaken. As requested, below we offer a detailed response to the specific concerns raised by you and by the reviewers. 

Reviewer 1

R1 overview: I had the pleasure of reading this interesting paper. As authors mentioned that “this is the largest and most comprehensive study of its kind”, I tend to agree that it is a large study of pandemic buying, both from the sample size point of view and also with regard to aspects considered, produced results are impressive. In my opinion, the paper did contribute to understanding why pandemic buying happens. However, there are points which in my opinion could further improve the paper

Response: Thank you for your feedback on our paper, we are very glad that you think it has the potential to contribute to this literature. We hope that we have adequately addressed your concerns about the paper, which we have responded to individually below. To give an overview of our changes, we have attempted to clarify the apparent contradiction that panic buying occurred yet only a small number of participants report engaging in high levels of this behaviour. We suggest that the stockouts observed at the beginning of the pandemic were driven by a number of people increasing their purchasing in modest amounts, rather than a minority engaging in extreme levels of this behaviour, which is supported by a recent report published by the IFS using barcode data from across the UK. In addition, we have included the papers that were recommended that have helped to justify our predictions and contextualise our findings in the discussion. 

R1. 1: Some parts have contradicting claims. In the result part, authors claimed that “Table 2 indicate that over-purchasing was reported by only a minority of both populations” (line 476). However, at the beginning of the paper, from line 56 to line 59, the author used one reference (1) to indicate that “panic buying” especially short of “toilet rolls” were happening, including in the UK. Authors provided explanation in the discussion why one nation is more caution than another, but did not clearly mention in the background whether panic buying was serious in UK or ROI. There is need to provide more evidence or report to agree or argue that the pandemic buying existed in UK along explanation to explain why the survey opposed the evidence.

Response: We appreciate this point needs clarifying and thank the reviewer for highlighting it. It is indeed the case that panic buying was observed in these nations, leading to total stockouts in some supermarkets. However, a recent report by the Institute for Fiscal Studies (https://www.ifs.org.uk/publications/15101) (which we now cite at the beginning of the introduction as well as in the discussion) found that the stockouts that were observed during the early stages of the COVID-19 pandemic were driven by a large number of people modestly increasing their purchases of tinned goods, toilet paper, rice etc. rather than a minority engaging in extreme levels of panic buying. As such, this can reconcile the apparent contradiction between the observation that panic buying occurred, but few people reported the highest levels of this behaviour. With regards to our point on line 476, we have attempted to clarify that panic buying occurred on a continuum to incorporate the fact that large numbers of the population increased their purchasing modestly and only few members of the population engaged in extreme levels of panic buying. This revised phrasing can be found in the first paragraph of the results section.

R1. 2: In the discussion part. When authors mentioned that “many were consistent with previous speculation about the psychological factors involved in this phenomenon”. It is needed to clarify what was consistent with which specific previous analysis? Currently, there is only have one reference for this statement.

Response: We have included a wider range of studies to support this statement and note which specific papers were supported by the findings of our analyses. Specifically, we note that our findings support previous research that has demonstrated anxiety and personality factors were associated with panic buying.

R1. 3: In discussion: “The country was particularly badly affected by the 2008 financial crisis but, probably more importantly, historical education in RoI places emphasis on the Great Famine of 1845-1849, which led to the death through starvation or disease of about one million people and the emigration of another million, resulting in a reduction in the Irish population of about 25% over just five years (69). As a consequence, scarcity cues may be more salient for people in Ireland compared to people in the UK”. This sound as a bold and subjective conclusion. I believe Britain also suffered from the pandemic in 1918/19, estimating 228,000 deaths. To conclude that RoI is more salient there is need to provide evidence and elaborate it.

Response: We had tried to consider historic reasons for differences between the UK and RoI in rates of panic buying but agree that this could be explained in more detail. However, we feel greater discussion of this topic would detract from the narrative of the paper and key findings regarding the psychological predictors of panic buying. On the recommendation of Reviewer 2 (R2. 15) we have decided to remove this paragraph from the discussion and elaborated on the importance of key variables in our model.

R1. 4: This question links back to my previous question, why the survey showed a minority of panic buying when it happened. Would this be related to “Personality” that authors mentioned? Is it possible that people are reluctant to admit they did “pandemic buying”?

Response: It is certainly true that people could be reluctant to admit that they engaged in ‘pandemic buying’ and we did find this in a pilot of the survey where participants were explicitly asked if they had engaged in this behaviour. However, our phrasing in the current study was more cautious to avoid this response bias. Furthermore, our finding that the minority of participants who engaged in panic buying is consistent with trends in the general population as demonstrated by the previously mentioned IFS report which is now referenced in both the introduction and the discussion. We observe a large number of people increasing their purchasing by modest amounts and only a minority who reported large increases in their purchasing behaviour. Nevertheless, we believe that these findings are still important in understanding the characteristics of individuals who engaged in panic buying, whether this was in modest or extreme amounts. We have included an additional line to explain this finding in the discussion on page 26 paraph 2, where we state: “This is consistent with evidence that stockouts during the early stages of the pandemic were driven by large numbers of consumers increasing their purchasing by modest amounts rather than a minority engaging in extreme levels of panic buying.”

R1. 5: There is need to reference some recent relevant work:

• Prentice, C., Chen, J., & Stantic, B. (2020). Timed intervention in COVID-19 and panic buying. Journal of Retailing and Consumer Services, 57, 102203.

• Zheng, Rui, Shou, Biying, Yang, Jun, 2020. Supply disruption management under consumer panic buying and social learning effects. Omega 102238

• ANZ Research, 2020. Panic buying or the essentials intensifies. Retrieved from. https ://www.savings.com.au/credit-cards/new-data-shows-massive-increases-and-d ecreases-in-consumer-spending-due-to-covid-19.

Response: Thank you for these paper recommendations! We have included the research you have recommended and have used Prentice et al. (2020) to help explain the finding that panic buying was observed more in the RoI compared to the UK, as RoI introduced their lockdown measures sooner than the UK, which was associated with panic buying in Prentice et al. This addition can be found on page 27, paragraph 2. 

In addition, we have used the findings of Zheng et al. (2020) to explain how social cues affect estimations of the foraging background rate. In the discussion, we use these findings to explain why neighbourhood belonging was positively associated with over-purchasing (contrary to our predictions), as those who were closer to their neighbours may have been more likely to observe or talk to peers about their purchasing habits. 

R1. 6: Why particularly to survey UK and RoI? What is the similarity/difference in these two specific countries that worth be compared? Would be better that authors compare the UK with a place that is not in Europe? This needs to be addressed or mentioned as limitation.

Response: The reasoning for selecting the UK and RoI was that these two nations shared enough economic, geographic (both being island nations), and political features to allow for our findings to be meaningfully compared across the countries. This allowed us to compare whether differences between approaches taken by the respective country’s Governments contributed to differences in over-purchasing, as this behaviour was observed to a greater degree in the RoI relative to the UK, which informed our discussion of these differences on page 27. 

However, we agree that it is important for future studies to consider whether these findings extend beyond Europe and in non-WEIRD samples to gain a better understanding of panic buying, as this phenomenon was observed globally. We have added this as a limitation of the present study, stating on page 31, paragraph 2: “Finally, it would be important for future studies to test whether these findings extend to countries outside of Europe, including the global South and in populations that are not WEIRD.“

R1. 7: Another aspect is with distribution of data, while it is mentioned that different age, sex, etc, has been considered, it would be also very usefull to show how panic buying was perceived in urban and regional areas. Other work presented as a one of the important factors.

Response: Thank you for this suggestion. We have included an ANOVA comparing urban, suburban, town and rural samples at the beginning of the results section. 

R1. 8: There should be a reference not a link, line 75.

Response: Thank you for highlighting this – we have amended this line to include a reference rather than a link.

Reviewer: 2

R2 Overview: The manuscript describes how several psychological factors are related consumers’ reported over-purchasing and panic buying in the early phase of COVID-19 pandemic in UK and Republic of Ireland. The study is based on representative surveys carried out at the end of March/ beginning of April with good sample sizes. The study is highly topical and addresses how individual responses vary in an exceptional situation using a wide battery of psychological measures as possible explanations.

Response: Thank you for your comments on our paper and are glad that it contributes towards this topical issue. We have addressed each of your comments below, providing additional detail about the correlations between the predictors included in the model.

R2. 1: Overall, the manuscript is clearly structured, but there are several long sentences that are difficult to follow as they contain several points that would benefit from separating them in two sentences (or other re-writing; e.g. lines 59-63, 95-100, 105-111, 127-131, 143-147, 155-159, 167-171, 201-206).

Response: We have revised the wording of the sentences you have highlighted and have revised any other long sentences we encountered through our revisions.

R2. 2: Although methods are described to some detail, especially the measures, the reporting raises some questions related to data quality that reader would like to know. What explains the large difference in completion times between the countries? What was done to check the data quality? Was the final data representative of the two countries? Did you have to exclude some responses due to too short response time?

Response: Thank you for raising this point and we appreciate there are some points that require additional clarification. With regards to differences in completion time, the RoI survey was launched slightly later than the UK survey and, as a result, some additional questions were added to it before it launched (e.g. whether people were currently abiding by the Irish lockdown rules and whether they expected to continue to do so, whether individuals come face to face with the public as part of their job, a measure of catastrophizing).

Extensive data quality checks were conducted on the data, including screening out individuals who did not provide full consent, lived outside the country of study, were under 18 years of age, completed in less than the minimum completion time and any potential duplicates were also removed. These data quality checks were done for both the UK and RoI data. For more information about checks to ensure the data quality, please see the protocol document for this research programme (reference 50). We have added a sentence to direct readers to this paper for information about the quality checks on page 14, paragraph 2.

Finally, the data was representative of the two countries, as the survey quotas were filled successfully. 

R2. 3: One of the strengths and weaknesses of the study are the many variables used to describe respondents’ living conditions, relation to COVID-19 pandemic, and psychological characteristics. However, there is very little information how these many independent variables relate to each other and the authors should provide a correlation matrix on this. One would expect distress, anxiety and depression symptoms to correlate positively with each other and with neuroticism, and further with the more domain specific measures, such as COVID 19 anxiety. The authors indicate that it would be interesting to study these relationships in the future, but that raises the question: why not do it with these data rather than collect new data or is the aim to use these same data in further analysis?

Response: We have produced a correlation matrix and this has now been added as supplementary material. We do intend to extend this line of investigation to develop more theoretically derived models that will attempt explain the associations among the predictor variables and how these are related to panic buying, and we will be conducting future survey waves which will address these issues amongst others. We believe that this is beyond the scope of this paper.

R2. 4: Currently the model contains variables that link very differently to over-purchasing. Why not run them in a hierarchical model to see how the different types of variables contribute to the overall explanation? Some are related to perceived infection status: although this may be a (very good) cause to over-purchase, it is an odd one to include in this kind of model as it does not relate to perception of oneself or living conditions in general. It relates to a short period of time which may have a very different impact whether the disease is on at the moment or has already been conquered. It would be important to know, how many people actually reported that they have had or thought to have had the disease: in the beginning of pandemic these cases are likely to be very low.

Response: This was a very interesting idea and the team devoted considerable time considering the possibility of a hierarchical model. However, we could not unanimously agree on the number of blocks, and the variables that would be in each block, that should be used to categorise the predictors. There were also very different views in terms of the potential ordering of the blocks to represent theoretical importance. However, the main problem was that the team did not feel that it was appropriate to state that we developed a post-hoc hierarchical model, as we would have been influenced by our knowledge of the results. 

R2. 5: The over-purchasing variable seems to have very skewed distribution. How was this dealt with in the regression analysis? Would it had been interesting to explain who are those 20% reporting to over-purchase?

Response: The parameters of the model were estimated using robust maximum likelihood, and this produces correct point estimates and standard errors under conditions of non-normality (Benson & Fleishman, 1994).

The analytic strategy was based on modelling the responses as a continuous latent dimension representing a propensity to over-purchasing, rather than identifying a cut-off to identify high levels of over-purchasing participants. The results from the regression analysis indicate those variables for which high scores would be associated with high levels of over-purchasing. 

See Benson, J., Fleishman, J.A. The robustness of maximum likelihood and distribution-free estimators to non-normality in confirmatory factor analysis. Qual Quant 28, 117–136 (1994). https://doi.org/10.1007/BF01102757

R2. 6: Conscientiousness being negatively related to over-purchasing may be linked to ability to think about the long-term consequences or it may simply be that respondents who were more conscientious were also that in following the official/governmental recommendations against hoarding.

Response: Thank you for this suggestion. We have added a small comment to this effect in the discussion.

R2. 7: Overall, the discussion does not go very deeply into the relative importance of the different variables in the strength of associations. In the limitations, the authors hint that looking at the relationships between variables would be of interest: perhaps the more general psychological variables mediate the relationships between COVID-19 based worries and over-purchasing. This would suggest that the current analysis of variables could go much further.

Response: We do not disagree with the notion that there is likely to a model underlying the predictor variables that may include mediated and moderated effects. We hope to be in a position to develop our theoretical model more fully in future, but at this stage we feel that it would be premature to propose and test any such model without adequate theoretical and empirical work to substantiate such a model.

R2. 8: Lines 197-199: Why is there is a direct link with children in the household and economic hardship and food security? Would this depend on the economic status of the household?

Response: Our modelling examined number of children and economic status as separate predictors of overpurchasing. Both were found to be significant as we had hypothesized.

R2. 9: Line 342: Heath problems should read Health problems

Response: We have amended this accordingly.

R2. 10: Lines 421-429: reliability of the IUS should be reported similar to other scales

Response: Thank you for highlighting this oversight! The reliability has now been entered.

R2. 11: Lines 494-495: This sentence should be included in Table 2 to make it self-explanatory

Response: We have included this in the table to ensure this is clear to readers.

R2. 12: Lines 507-512: This would be better placed in data analysis part as a conclusion to explaining the factor analyses

Response: We have moved this section accordingly and agree this presentation is clearer.

R2. 13: Lines 542-543: referring to earlier findings: move to discussion

Response: We have now removed the reference to previous research in the results section and refer to this in the discussion. 

R2. 14: Table 3: are the reported R2 values referring to adjusted values; there is an additional column in the Table on page 27. Make the Table self-explanatory: write out CRT

Response: Adjusted R-squared values have been added. The CRT entry has been amended to read “Cognitive Reflection Task”.

R2. 15: Discussion, page 29: referring to the Great Famine in Ireland in mid 1800 as a possible explanation for country differences sounds a bit far-fetched. How can one align the situation in Ireland now vs then? Why not other events or World War 2 experiences? I suggest to omit this.

Response: We have followed your recommendation and omitted this paragraph from the text. Rather, we elaborate on the importance of different variables and the strength of their associations as you note in your comment above. We hope that this presents a clearer discussion of our findings and the importance of considering these variables to avoid future instances of panic buying.

Reviewer: 3

R3 Overview: While the authors are to be commended on introducing the biological theory, I find its application not convincing: They write that “In the simple context of supermarket purchases, this is the choice between buying goods from a local supermarket, which has an observable distribution of goods (e.g. canned goods, dried food), or sacrificing the cost in time and effort required to travel to a more distant supermarket where the distribution of these goods is unknown.” I do not think this is true because a consumer will likely find the same supermarkets even when shopping elsewhere, being equipped with the very same products. It is quite a stretch to propose that this is ‘unknown’ to the consumer. As this is a key theoretical framework, I find it problematic that the argument is not convincingly applied to consumer realities.

Response: We would like to thank the reviewer for their comments and appreciate their reservation about the application of foraging theory. We have revised the paper to explain in greater detail how foraging theory can explain consumer behaviour. In doing so, we refer to a paper that has been published since our initial submission by Dickins and Schaltz (2020) that also applies foraging theory to panic buying during the COVID-19 pandemic. 

With regards to the statement about supermarket goods that was highlighted, we have clarified that it is not the variety of products that is unknown, but rather the abundance of these products in other supermarkets. In foraging theory, this is known as the ‘background rate’ of resources, which is the individual’s estimation of the resources that are available in patches (or supermarkets) in the environment that are within travelling distance from their current patch (or supermarket). As such, this information is not directly available to the consumer but is inferred from cues (such as scarcity cues) in their environment. 

R3.1: Further, I am not sure whether the theoretical lens is sufficient to motivate the hypotheses and included variables (also see below). I would also suggest that the authors broaden their theoretical discussions and include the literature on consumer behaviour on scarcity effects more (there is a lot, e.g.: Hamilton, R., Thompson, D., Bone, S., Chaplin, L. N., Griskevicius, V., Goldsmith, K., ... & Piff, P. (2019). The effects of scarcity on consumer decision journeys. Journal of the Academy of Marketing Science, 47(3), 532-550.)

Response: Thank you for your suggestion of including this paper. We have integrated the content you have suggested on scarcity effects on consumer behaviour within our theoretical section and explain how foraging theory can complement the findings of research investigating scarcity cues. We include this additional content on pages 8-9, stating: “The application of foraging theory to this context complements work that has examined the effects of manipulating product availability on consumers’ behaviour. One common and effective marketing method that has been used to increase the demand for products is by manipulating scarcity cues, which indicate the availability of items (Hamiton et al., 2019). Foraging theory presents a lens through which to understand the efficacy of such marketing tactic; through presenting cues that the environment is poorer in resources (i.e. fewer items are available), the forager should infer a lower background rate and hence forage the current patch more extensively (17). Notably, it has been observed that it is the popularity of items, rather than their exclusivity that drive scarcity effects, suggesting this effect relies on the behaviour of other consumers rather than retailers (Parker & Lehmann, 2011). This is consistent with evidence that observing others engage in panic buying was positively associated with increased consumer’s own panic buying behaviours during the COVID-19 pandemic (Zheng et al., 2020). As such, other consumers are prominent indicators of scarcity and thus the perception of a fall in the background rate.”

R3.2: A weakness of the current paper is that the interesting front-end (using foraging theory) has not logical links with the hypotheses and empirical design of the study. Such a gap is problematic. It is not clear how the introduced and applied theory helps to develop the hypotheses. Clearly, the theoretical foundation is a biological one, but the hypotheses refer to household income or personality dimensions. I am not saying that this cannot be derived from the theory but the authors just make to little effort to link these parts. This needs significant revision.

Response: We have revised our introduction and theoretical section to make the links to the hypotheses clearer. Specifically, we highlight that certain psychological and demographic variables may increase the salience of scarcity cues, which influence the perception of the background rate of resources depleting (in line with foraging theory). These factors should therefore stimulate greater over-purchasing behaviour to avoid resource scarcity. We hypothesise that there are variables that increase the salience of scarcity cues and should therefore be associated with over-purchasing, which have guided our predictions. 

We have also expanded our theoretical section to introduce the concept of ‘risk of predation’ by Dickins and Schaltz. This refers to the individual’s perception of their risk of infection through coming into close contact with other individuals. Individuals who perceive this risk to be high should hoard items to reduce the number of shopping trips required and therefore their risk of infection. This additional content can be found on pages 9-10: “A further parameter that affects foraging behaviour is the risk of predation. While viruses are parasites, rather than predators, the nature by which the virus spreads, i.e. through close contact with other human beings, should increase vigilance and subsequently lead to other people becoming associated with a threat to life (Dickins & Schaltz, 2020). In order reduce exposure to infection, individuals who perceive a high risk of infection should increase their foraging effort to hoard a greater amount of resources to maximise inter-foraging delays. This strategy minimises the frequency of encounters that put the individual at risk. Therefore, along with tracking scarcity cues, perceived risk of infection should predict panic buying behaviour.” 

This content gives additional context to some of our predictions and we hope we have sufficiently motivated the hypotheses using a foraging framework through our revisions. 

R3.3: Also, the authors need to develop each individual hypothesis in more detail. They include various interesting variables, I am not doubting the suitability. However, the selection seems abitrary.

Response: 

We appreciate that the referee does not entirely ‘buy’ out foraging-based model, but we think we have explained how the variables we have selected fit within that framework and have, indeed, provided a diagrammatic representation which we think makes this clear. We have a lot of variables and adding substantially to what we have already written will result, we think, in a paper that is long and difficult to read.

R3.4: Related to this, I really miss a narrative of this research: It appears that a lot of variables have been tested without sufficient theoretical reasoning and a guiding theoretical framework. The paper would benefit significantly from a sharper profile.

Response: 

We disagree about this and think that the scientific community should be allowed to make their own judgment on this issue.

R3.5: As for the method, have the authors controlled for common method bias? And other potentially biasing response behavior?

Response: Unfortunately we could not control for common method bias as all the data were from self-report methods. We have added a sentence to the limitations section that now reads-

“In future studies, it will be useful to refine our model to explicitly address the inter-relationships between the variables and the possibility that some (for example, the anxiety measures) are tapping a common latent process. Also, methods to avoid potential bias from the use of common methods (self-report) will be considered.”

R3.6: Also, can the authors reason their choice of method, e.g., why a SEM has not been used.

Response: SEM, or latent variable modelling, was used. To make this clearer, we have added text in the Methods-

“All of these analyses were conducted using latent variable modelling in Mplus 8.1 (67).” 

R3.7: Overall, the authors draw on a creative theoretical basis but there is a logic gap between this section and the empirical design. This is problematic but may be alleviated through a thorough revision. Good luck.

Response: We thank you for your comments and hope that through our revisions we have made closer ties between our theoretical background and empirical design. We have explained how foraging theory is an important framework for understanding consumer behaviour, which can complement research on the effect of scarcity cues on consumers’ purchasing behaviour. In addition, we highlight how the variables included in the present study influence consumers’ estimation of resources in the environment (the background rate), which motivates over-purchasing behaviours. Our design was able to test the hypothesised associations between the variables we included and panic buying. However, we acknowledge that we did not include direct measures of scarcity or perceptions of the background rate of resources in our limitations, which would be an important avenue for future research to investigate. We also acknowledge that foraging theory is a framework which you will still likely find unconvincing but think that, with the changes you have prompted, we have now clarified it to the point where it should be judged by the research community.

Finally, please note that we have added Sarah Butter to the author list. Sarah joined our project after the data collection but before we made the original submission. Although we omitted her name initially this was an oversight and, on reflection, we think she definitely deserves to be included because she did quite a lot of work critiquing and suggesting changes to the original manuscript and has also worked with me on the revised version.

We have made a serious effort to address all of the points raised in the review of our manuscript and hope that our revised version is now suitable for PLOS ONE. Many thanks for reconsidering this paper for publication. 

Yours with best wishes,

Richard Bentall PhD FBA

---

## [Editor Report · Decision Letter 1]

18 Jan 2021

Pandemic buying: Testing a psychological model of over-purchasing and panic buying using data from the United Kingdom and the Republic of Ireland during the early phase of the COVID-19 pandemic

PONE-D-20-25983R1

Dear Dr. Bentall,

We’re pleased to inform you that your manuscript has been judged scientifically suitable for publication and will be formally accepted for publication once it meets all outstanding technical requirements.

Kind regards,

Frantisek Sudzina

Academic Editor

PLOS ONE
---

## [Editor Report · Acceptance letter]

22 Jan 2021

PONE-D-20-25983R1 

Pandemic buying: Testing a psychological model of over-purchasing and panic buying  using data from the United Kingdom and the Republic of Ireland during the early phase of the COVID-19 pandemic 

Dear Dr. Bentall:

I'm pleased to inform you that your manuscript has been deemed suitable for publication in PLOS ONE. Congratulations! Your manuscript is now with our production department. 

Kind regards, 

on behalf of

Dr. Frantisek Sudzina 

Academic Editor

PLOS ONE